

# A Chern-Simons theory for dipole symmetry

Xiaoyang Huang⋆

Department of Physics and Center for Theory of Quantum Matter,
University of Colorado, Boulder CO 80309, USA

⋆ xiaoyang.huang@colorado.edu

## Abstract

We present effective field theories for dipole symmetric topological matters that can be described by the Chern-Simons theory. Unlike most studies using higher-rank gauge theory, we develop a framework with both $U(1)$ and dipole gauge fields. As a result, only the highest multipole symmetry can support the 't Hooft anomaly. We show that with appropriate point group symmetries, the dipolar Chern-Simons theory can exist in any dimension and, moreover, the bulk-edge correspondence can depend on the boundary. As two applications, we draw an analogy between the dipole anomaly and the torsional anomaly and generalize particle-vortex duality to dipole phase transitions. All of the above are in the flat spacetime limit, but our framework is able to systematically couple dipole symmetry to curved spacetime. Based on that, we give a proposal about anomalous dipole hydrodynamics. Moreover, we show that the fracton-elasticity duality arises naturally from a non-abelian Chern-Simons theory in 3D.

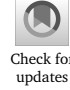

# 1 Introduction

In recent years, there has been increasing interest among condensed matter and high energy physicists in one family of novel phases of matter that is characterized by fractons – excitations with restricted mobility [1–5]. As a simple example of the restricted mobility, a system with both charge/mass and dipole moment/center of mass conservation forbids single-particle dynamics and only allows the charge to move by inserting a dipole moment. Such kinetic constraint results in a fruitful unconventional phase of matter, including ergodicity breaking [6,7], dipole condensation [8–12], and (breakdown of) dipole hydrodynamics [13–21].[1] Experiment [22] shows strong evidence about several above features making the universality class reliable. The study of fractons also triggers a new type of symmetry, the subsystem symmetry, that only acts on a sub-dimensional manifold of the whole spacetime [23,24] (see also a review [25]). A continuum quantum field theory for fracton has been developed showing exotic features like exponentially large ground-state degeneracy and UV/IR-mixing [26].

Inspired by a series of seminal works of Michael Pretko [27–29], a higher-rank gauge theory has been developed to describe fractonic matters [30] and is used to generalize the topological Chern-Simons field theory [31–33]. However, the higher-rank Chern-Simons theory has two major disadvantages. First, this theory requires substantial insight to write down a gauge invariant action,[2] and lacks a systematic way to classify what kind of Chern-Simons terms that can be written down based simply on the symmetries we have. Second, this theory is unable to couple to curved spacetime consistently [34,35], and it is unclear as to how to characterize the deviation from a gauge invariant theory on the curved spacetime, although progress has been made recently [19,36,37]. The ability to define a Chern-Simons theory on a generic spacetime reflects its topological nature, which seems to be lacking in the higher-rank Chern-Simons theory.

In this paper, we want to study the interplay between fracton-like multipole symmetry [38] and topology by developing a Chern-Simons theory for both the $U(1)$ and dipole gauge fields: $(A_\mu, A_\mu^a)$, following [19]. Before moving on, let us clarify our notations. We denote $\mu, \nu = t, x, y, z, \dots$ for the physical spacetime, $\alpha, \beta = t, x, y, z, \dots$ for the internal spacetime, and use $i, j$ and $a, b$ to indicate their spatial subspace, respectively. The $n$-multipole gauge field can be written as $A_\mu^{a_1 \cdots a_n}$, but, for our purpose, we restrict our attention to dipole symmetry. The set of gauge fields for dipole symmetry is analogous to scalar-and-vector charge theory [39,40], and the internal index indicates whether it behaves as a scalar or vector under rotational symmetry. Diplole symmetry further requires a nontrivial coupling between scalar and vector charges. Under $U(1)$ ($\alpha$) and dipole ($\xi^a$) gauge transformations, the two gauge fields transform as, in the flat spacetime limit,

$$A_\mu(t, x) \to A_\mu(t, x) + \partial_\mu \alpha(t, x) + \delta_{\mu a} \xi^a(t, x), \tag{1a}$$

$$A_\mu^a(t, x) \to A_\mu^a(t, x) + \partial_\mu \xi^a(t, x), \tag{1b}$$

---

[1]One might notice that some of them are hydrodynamics of dipole superfluid. This is because, when momentum conservation is present, dipole symmetry must be broken [19].

[2]If it is reduced from a $\theta$-term [29] in one-higher dimension, the construction of the $\theta$-term is still nontrivial.

and we see that $A_\mu$ transform nontrivially under the dipole shift $\xi^a$. By setting $\xi^a = -\delta^{ia}\partial_i\alpha$, they combine to form a symmetric higher-rank gauge theory with $(A_t, A_{ij})$, and whose gauge transformation reads $A_t \to A_t + \partial_t\alpha$, $A_{ij} \to A_{ij} - \partial_i\partial_j\alpha$. However, we emphasize that it is *unnecessary* to reduce the dipole gauge theory to the higher-rank gauge theory for the latter overshadows many properties of the original dipole gauge theory for several reasons. For example, first, the scalar and vector charges are mixed under higher-rank gauge theory and combined to form an effective symmetric tensor charge. This change of the underlying degrees of freedom is an unwanted feature for the purpose of constructing an effective field theory as their transformations under rotational symmetry are altered.[3] Second, the higher-rank gauge fields are unable to be expressed as differential forms. Unlike it, our set of gauge fields is manifestly invariant under diffeomorphism and can be expressed as 1-forms: $A = A_\mu dx^\mu$, $A^a = A^a_\mu dx^\mu$. Now, recall that the conventional Chern-Simons theory is built upon differential forms of gauge fields for a scalar charge, our dipole gauge theory is thus better suited to generalize the Chern-Simons theory than the high-rank gauge theory.

In Section 2, we discuss the dipolar Chern-Simons theory in even spatial dimensions and its boundary 't Hooft anomaly. Using the dipole gauge fields, we can write down most generally in $D = 2n + 1$-spacetime a dipolar Chern-Simons theory

$$S_{CS} = C_{2n} \int d^D x \; \epsilon^{\mu_1 \nu_1 \mu_2 \cdots \nu_n \mu_{n+1}} A^{a_1}_{\mu_1} \partial_{\nu_1} A^{a_2}_{\mu_2} \cdots \partial_{\nu_n} A^{a_{n+1}}_{\mu_{n+1}} f_{a_1 a_2 \cdots a_{n+1}}, \tag{2}$$

where $f_{a_1 a_2 \cdots a_{n+1}}$ is an invariant tensor for the underlying (discrete) rotational symmetry. The coefficient $C_{2n}$ is shown in Appendix A to be quantized for a compact dipole symmetry. From construction, the Chern-Simons term is guaranteed to be invariant under rotational symmetry, and invariant under gauge transformations up to a total derivative term. Interestingly, we find the dipolar Chern-Simons theory can also exist in odd spatial dimensions (see [41]), which is impossible for conventional scalar charge Chern-Simons theory. In Section 3, we detail such construction and study its boundary anomaly. Due to breaking continuous rotational symmetry to some discrete subgroups, we show how the boundary anomaly depends on the direction of the boundary. This will provide a novel example where bulk-edge correspondence displays a dependence on the boundary itself.

In Section 4, we show that the torsional anomaly in $U(1)$ quantum Hall state can be identified as a dipole-like anomaly. Intuitively, momentum is like a (time-reversal odd) vector charge, so it shares many similarities with dipole symmetry. The torsional anomaly provides a mechanism to generate gapless modes in the quantum Hall state.

Conventional particle-vortex duality is captured by a mixed Chern-Simons term between dynamical and background gauge fields. In Section 5, we generalize it to dipole phase transitions that were reported recently in [42]. We study the mixed Chern-Simons term that incorporates either a dipole symmetry or a dipole symmetry breaking and show how the Lifshitz theory emerges from it.

All of the above are defined in the flat spacetime limit. In Section 6, we develop effective field theories toward a curved spacetime dipolar Chern-Simons theory. In Section 6.1, we generalize (2) to curved spacetime with the help of dipole Goldstone. We then move further to consider a pure gauge theory in 3D spacetime in Section 6.2. By treating the dipole symmetry in an equal footing as the spacetime symmetry, we arrive at a non-abelian Chern-Simons theory following the canonical construction of topological 3D gravity [43–46]. The resulting theory is a generalized Wen-Zee term [47], and we will show that it enriches present understandings of "fracton-elasticity duality" [48–50] (see also a review [51]).

---

[3]For example, it was shown in [19] that the dipole current in dipole hydrodynamics will have anti-symmetric components that must couple to the anti-symmetric part of the dipole gauge field. Such coupling is disallowed in the symmetric higher-rank gauge theory.

## 2 Dipole anomaly in one spatial dimension

We start by proposing a boundary anomalous theory and then search for a bulk Chern-Simons theory that cancels the boundary anomaly.

Denote the compact phase variables for charge and dipole moment as $\phi, \phi^x$. In $d = 1$, $\phi^x$ is like a scalar charge. Under $U(1)$ and dipole symmetry transformations, they shift by

$$\phi(x) \to \phi(x) + \alpha - x\xi^x \quad (\mathrm{mod}\ 2\pi), \tag{3a}$$

$$\phi^x(x) \to \phi^x(x) + \xi^x. \tag{3b}$$

Consider a $D = 1 + 1$ system described by a real-time action that is invariant under the above global symmetry

$$S[\phi, \phi^x] = \int \mathrm{d}^2x\ \frac{\chi}{2}(\partial_t\phi)^2 - \frac{K_n}{2}(\partial_x\phi + \phi^x)^2 - C\partial_t\phi^x\partial_x\phi^x + \frac{\kappa}{2}(\partial_t\phi^x)^2. \tag{4}$$

In the action, we did not include the kinetic part $(\partial_i\phi^x)^2$ as the system is not in the symmetry broken phase; see Section 5 for dipole symmetry breaking. The $C$-term, which has the first-order time derivative, will be responsible for the anomaly in the system. The Noether current is obtained by allowing $\alpha(t, x)$ and $\xi^x(t, x)$ to be spacetime dependent. This leads to, to the leading order in $\alpha, \xi^x$,

$$\delta S = \int \mathrm{d}^2x\ \chi\partial_t\phi(\partial_t\alpha - x\partial_t\xi^x) - K_n(\partial_x\phi + \phi^x)(\partial_x\alpha - x\partial_x\xi^x) - 2C\partial_t\phi^x\partial_x\xi^x + \kappa\partial_t\phi^x\partial_t\xi^x$$

$$= \int \mathrm{d}^2x\ J^\mu\partial_\mu\alpha + \hat{J}_x^\mu\partial_\mu\xi^x, \tag{5}$$

from which we identify the currents as

$$J^t \equiv n = \chi\partial_t\phi, \quad J^x = -K_n(\partial_x\phi + \phi^x), \quad \hat{J}_x^t = \kappa\partial_t\phi_x - xn, \quad \hat{J}_x^x = -2C\partial_t\phi_x - xJ^x. \tag{6}$$

Notice that the dipole current $\hat{J}_x^\mu$ has a component proportional to $xJ^\mu$ that describes the orbital part of the dipole moment. Now, we try to gauge the action (4) by adding the corresponding gauge fields $\hat{A}_\mu, A_\mu^x$. The gauged action is given by

$$S[\phi, \phi^x, A, A^x] = S[\phi, \phi^x] - \int \mathrm{d}^2x\ (J^\mu\hat{A}_\mu + \hat{J}_x^\mu A_\mu^x)$$

$$+ \int \mathrm{d}^2x\ \left(\frac{1}{2}A_t^x(\kappa A_t^x - 2CA_x^x) + \frac{\chi}{2}(\hat{A}_t - xA_t^x)^2 - \frac{K_n}{2}(\hat{A}_x - xA_x^x)^2\right), \tag{7}$$

where the second line is some suitable counterterm. Under gauge transformations $\phi \to \phi + \alpha - x\xi^x$, $\phi^x \to \phi^x + \xi^x$ and $\hat{A}_\mu \to \hat{A}_\mu + \partial_\mu\alpha$, $A_\mu^x \to A_\mu^x + \partial_\mu\xi^x$, the gauged action changes by

$$\delta S[A, A^x; \alpha, \xi^x] = \int \mathrm{d}^2x\ C\xi^x(\partial_x A_t^x - \partial_t A_x^x). \tag{8}$$

It is impossible to completely remove this anomalous gauge transformation by adding further local counterterms, therefore, (4) encounters an 't Hooft anomaly. The 't Hooft anomaly can be canceled by a bulk Chern-Simons theory. Consider the Chern-Simons term in $D = 2 + 1$[4]

$$S_{CS}[A^x] = C\int \mathrm{d}^3x\ \epsilon^{\mu\nu\rho}A_\mu^x\partial_\nu A_\rho^x. \tag{9}$$

---

[4]We are informed that Leo Radzihovsky has constructed a similar Chern-Simons term for vector charge theory in an unpublished work.

Suppose the action is only defined on $y \leq 0$. Under a gauge transformation, the Chern-Simons action changes by

$$\delta S_{CS}[A^x; \xi^x] = C \int \mathrm{d}^3x \; \epsilon^{\mu\nu\rho} \partial_\mu(\xi^x \partial_\nu A^x_\rho)$$

$$= C \int \mathrm{d}^2x \; \xi^x \epsilon^{\nu\rho} \partial_\nu A^x_\rho = -\delta S \,. \tag{10}$$

Hence, a gauge invariant theory is a sum of the Chern-Simons action (9) and the boundary action (4).

The dipolar Chern-Simons theory in (9) is anisotropic, i.e. it does not involve the $y$-dipole. If, instead, we have an isotropic dipolar Chern-Simons theory

$$S_{CS}[A^a] = C \int \mathrm{d}^3x \; \epsilon^{\mu\nu\rho} A^a_\mu \partial_\nu A^a_\rho \,, \tag{11}$$

for $a = x, y$, its boundary must also be anomalous in terms of the $y$-dipole moment $\phi^y$. However, since we put the boundary at $y = 0$, the $\phi^y$ is out of the plane. This implies that if looking at the line $y = 0$, $\phi^y$ would move like an unconstrained charge since its dipole moment can always be preserved. Therefore, the 't Hooft anomaly for the $y$-dipole is identical to that for $U(1)$ charge without dipole symmetry. Nevertheless, this out-of-plane dipole anomaly will be important for the discussion in Section 3.

The Chern-Simons theory (11) has appeared in [31] in a different manner. They used higher-rank gauge fields $(A_t, A_{ij})$ to construct it as

$$S_{gCS}[A_{ij}, A_t] = C \int \mathrm{d}^3x \left( \epsilon^{ij} \dot{A}_{ik} A^k_j - 2A_t \epsilon^{ij} \partial_i \partial_k A^k_j \right) . \tag{12}$$

By taking $A^a_t = \partial_a A_t$ and identifying the physical spacetime with the internal spacetime, we find (11) reduces to (12). This is consistent with the gauge fixing discussed below (1) in order to reduce the dipole gauge fields to the higher-rank gauge fields. Several remarks follow. First, the way the higher-rank gauge theory is used to construct (12) does not have a direct generalization to our most general Chern-Simons theory in (2). Second, the length $(L)$ dimension of each field is different in the two theories. For (12), they have $[A_{ij}] = L^{-1}$ and $[A_t] = L^0$ [31]. However, we have $[A^a_\mu] = [A_\mu] = L^{-1}$ and the dimensional analysis is taken with respect to the physical spacetime only. The latter principle further indicates that $[\delta^a_\mu] = L^{-1}$ and $[x^a] = L^0$ because $e^a_\mu = \delta^a_\mu$ is a 1-form field and $x^a$ lives in the internal spacetime.

The action variation (5) leads to a conservation of dipole current as $\partial_\mu \hat{J}^\mu_x = 0$. This is not written in the canonical way where dipole current is not conserved (see [19]), and this is because $\hat{J}^\mu_x$ contains the orbital dipoles. To have the non-conservation of dipole current, we define the intrinsic dipole current $J^\mu_x = \hat{J}^\mu_x + xJ^\mu$, and then the dipole Ward identity changes to $\partial_\mu J^\mu_x = J^x$. Along with it is the change of the $U(1)$ gauge field $A_\mu = \hat{A}_\mu - xA^x_\mu$, which then transforms as

$$A_\mu \to A_\mu + \partial_\mu \alpha - x\partial_\mu \xi^x = A_\mu + \partial_\mu \alpha' + \delta_{\mu x} \xi^x \,. \tag{13}$$

This agrees with (1). Now, combining (5) and (10), we obtain the anomalous equations of motion

$$\partial_\mu J^\mu = 0 \,, \tag{14a}$$

$$\partial_\mu J^\mu_x = J^x + C\epsilon^{\mu\nu} F^x_{\mu\nu} \,, \tag{14b}$$

where $F^x_{\mu\nu} = \partial_\mu A^x_\nu - \partial_\nu A^x_\mu$.

The boson action (4) indicates that $\phi^x$ is gapped [42]: By the change of variable $\phi^{x\prime} = \phi^x + \partial_x\phi$, the $K_n$-term generates a mass term for $\phi^{x\prime}$, so we can set $\phi^{x\prime} = 0$, and obtain (up to higher derivative corrections)

$$\phi^x \approx -\partial_x\phi. \tag{15}$$

This is also clear from the equation of motion (14b) that in the absence of the external field, we have $J^x \approx O(\partial^2\phi_x)$, so using (6), we find (15) to the leading derivative order. Now, keeping the next-leading derivative order, we have $J^x \approx \partial_x J_x^x \approx 2C\partial_t\partial_x^2\phi$. Using $\partial_\mu J^\mu = 0$, we arrive at the chiral equation

$$\chi\partial_t\phi + 2C\partial_x^3\phi = 0. \tag{16}$$

Upon Fourier transformation, we obtain a cubic chiral mode

$$\omega = -\frac{2C}{\chi}k_x^3. \tag{17}$$

It was recognized that the damped anomalous chiral mode in $d = 1$ will flow to the Kardar-Parisi-Zhang (KPZ) universality class [52]. To study dissipation, one needs to construct an effective field theory on a Schwinger-Keldysh contour. This has been done recently in [15] though for different purposes. According to [15], the cubic chiral mode would experience a quartic dissipation forming a damped mode $\omega \sim k_x^3 - ik_x^4$. However, nonlinearity is relevant in $d = 1$. By a zeroth-order scaling analysis, the true dissipative fixed point is predicted to be $\omega \sim k_x^3 - ik_x^z$ with $z \approx 7/2$, which, a priori, does not belong to the KPZ class [15]. Therefore, our dipolar Chern-Simons theory provides a concrete example to realize the novel universality class beyond KPZ.

In the presence of dipole symmetry, the $U(1)$ chiral anomaly is forbidden. This can be seen through both boundary and bulk theories. The boundary chiral boson action for $U(1)$ anomaly must take $\partial_t\phi(\partial_x\phi + \phi^x)$ to preserve the dipole symmetry. According to (15), the anomalous charge flow is gapped and charges cannot propagate by themselves, so such a term gives trivial dynamics at the boundary. From the perspective of the bulk, there is no gauge invariant $U(1)$ Chern-Simons theory under gauge transformation (1). For example, if $U(1)$ gauge transformation is preserved, we can have $\int \epsilon^{\mu\nu\rho}A_\mu\partial_\nu A_\rho$, but it is not dipole gauge invariant. If considering a mixed gauge interaction $\int \epsilon^{\mu\nu\rho}(A_\mu\partial_\nu A_\rho - 2A_\mu\delta_{\nu x}A_\rho^x)$, it fixes the dipole gauge transformations of the first term but, at the same time, generates more terms under both $U(1)$ and dipole gauge transformations. This fact can already be generalized to systems that preserve multipole symmetries, and in that case, *only* the highest multipole symmetry can support the 't Hooft anomaly.

## 3 Dipole anomaly in two spatial dimensions

Since $U(1)$ Chern-Simons theory only exists in even spatial dimensions, $U(1)$ anomaly can only occur in odd spatial dimensions. This is no longer true for dipolar Chern-Simons theory and dipole anomaly together with an appropriate discrete rotational symmetry. To see it, consider $D = 3 + 1$ spacetime, and introduce the vielbein $e_\mu^a$. Here, $e_\mu^a$ transforms as a vector under rotational symmetry. It is important to work in flat spacetime $e_\mu^a = \delta_\mu^a$ such that the vielbein is not a truly gauge field; we will come back to gauging both dipole and spacetime in Section 6.2. Combining the two 1-forms $\delta_\mu^a$ and $A_\mu^a$, we can construct the following Chern-Simons theory

$$S_{CS} = C\int d^4x\, \epsilon^{\mu\nu\rho\sigma}A_\mu^a\partial_\nu A_\rho^b\delta_\sigma^c f_{abc}, \tag{18}$$

where $f_{abc}$ is a totally symmetric invariant tensor under a discrete rotational symmetry. In the following, we will consider two different point group symmetries, construct bulk dipolar Chern-Simons theories, and study their corresponding boundary anomalies. Note, this perspective is the inverse of that taken in Section 2.

## 3.1 Tetrahedral dipole anomaly

Consider a three-dimensional fluid with tetrahedral symmetry [53]. There exists a dipolar Chern-Simons term

$$S_{CS} = C_{\mathrm{T}} \int \mathrm{d}^4 x \; \epsilon^{\mu\nu\rho\sigma} A_\mu^a \partial_\nu A_\rho^b \delta_\sigma^c f_{abc}^{\mathrm{T}} \,, \tag{19}$$

with $f_{xyz}^{\mathrm{T}} = f_{xzy}^{\mathrm{T}} = f_{yzx}^{\mathrm{T}} = f_{zxy}^{\mathrm{T}} = 1$. While the dipolar Chern-Simons term breaks time-reversal symmetry, it does not break parity in all three directions. By contrast, a $U(1)$ Chern-Simons term breaks parity in every direction. Suppose putting boundary at $z = 0$, the dipolar Chern-Simons theory under gauge transformation changes by

$$\delta S_{CS}(z=0) = C_{\mathrm{T}} \int \mathrm{d}t\mathrm{d}x\mathrm{d}y\mathrm{d}z \; \epsilon^{\mu\nu\rho\sigma} \partial_\mu (\xi^a \partial_\nu A_\rho^b \delta_\sigma^c f_{abc}^{\mathrm{T}}) = C_{\mathrm{T}} \int \mathrm{d}t\mathrm{d}x\mathrm{d}y \; \epsilon^{\nu\rho I} \xi^a \partial_\nu A_\rho^b f_{abI}^{\mathrm{T}} \,, \tag{20}$$

where capital letters $I, J$ run over the two-dimensional boundary. Manipulating in a reverse way as Section 2, we find the chiral boson action

$$S[\phi, \phi^a] = \int \mathrm{d}t\mathrm{d}x\mathrm{d}y \; \frac{\chi}{2}(\partial_t \phi)^2 - \frac{K_n}{2}(\partial_I \phi + \phi_I)^2 - C_{\mathrm{T}}\epsilon^{IJ}\partial_t \phi^a \partial_I \phi^b f_{abJ}^{\mathrm{T}} + \frac{\kappa}{2}(\partial_t \phi^a)^2 \,. \tag{21}$$

By varying the action with respect to the boson fields and using $\phi_I \approx -\partial_I \phi$, we arrive at the equations of motion

$$\chi \partial_t \phi - 2C_{\mathrm{T}}(\partial_x^2 - \partial_y^2)\phi_z = 0 \,,$$
$$\kappa \partial_t \phi_z + 2C_{\mathrm{T}}(\partial_x^2 - \partial_y^2)\phi = 0 \,. \tag{22}$$

Upon Fourier transformation, it leads to two modes

$$\omega = \pm \sqrt{\frac{4C_{\mathrm{T}}^2}{\kappa\chi}} |k_x^2 - k_y^2| \,. \tag{23}$$

The two counterpropagating modes form a time-reversal pair. Since the tetrahedral group is symmetric in exchanging $x, y, z$, the boundary anomaly at $x = 0$ or $y = 0$ will be similar. However, the anomaly does differ if the boundary is located in an arbitrary direction. To see it, we rotate the invariant tensor through $f_{acd}^{\mathrm{T,R}} = R_{ab}^y f_{bcd}^{\mathrm{T}}$, where $R^y$ is the rotation matrix along $y$. Parametrizing the rotation matrix by $\theta$, we have

$$f_{zxy}^{\mathrm{T,R}} = f_{zyx}^{\mathrm{T,R}} = f_{xyz}^{\mathrm{T,R}} = f_{xzy}^{\mathrm{T,R}} = \cos\theta \,,$$
$$f_{xxy}^{\mathrm{T,R}} = f_{xyx}^{\mathrm{T,R}} = -f_{zyz}^{\mathrm{T,R}} = -f_{zzy}^{\mathrm{T,R}} = \sin\theta \,,$$
$$f_{yxz}^{\mathrm{T,R}} = f_{yzx}^{\mathrm{T,R}} = 1 \,. \tag{24}$$

Then, the chiral boson action at $z = 0$ is given by

$$S[\phi, \phi^a] = \int \mathrm{d}t\mathrm{d}x\mathrm{d}y \; \frac{\chi}{2}(\partial_t \phi)^2 - \frac{K_n}{2}(\partial_I \phi + \phi_I)^2 - C_{\mathrm{T}}\epsilon^{IJ}\partial_t \phi^a \partial_I \phi^b f_{abJ}^{\mathrm{T,R}} + \frac{\kappa}{2}(\partial_t \phi^a)^2 \,. \tag{25}$$

Varying the action, we obtain

$$\chi \partial_t \phi - C_T(2\cos\theta\, \partial_x^2 - (\cos\theta + 1)\partial_y^2)\phi_z + 2C_T \sin\theta(\partial_x^3 - \partial_x\partial_y^2)\phi = 0,$$
$$\kappa \partial_t \phi_z + C_T(2\cos\theta\, \partial_x^2 - (\cos\theta + 1)\partial_y^2)\phi + 2C_T \sin\theta\, \partial_x\phi_z = 0. \tag{26}$$

When $\theta = 0$, we recover (23). However, as soon as $\theta \neq 0$, the boundary modes changes, to the leading order in wavevectors, as

$$\omega \approx \frac{2C_T}{\chi}\left(-\frac{k_x^3}{\sin\theta} + k_x k_y^2 \cot\frac{\theta}{2}\right), \tag{27a}$$

$$\omega \approx \frac{2C_T \sin\theta}{\kappa} k_z. \tag{27b}$$

The two modes are now chiral and break the time-reversal symmetry as well as the parity of $x$ and $z$, but preserve the parity of $y$. This is consistent with the rotated invariant tensor, or equivalently the rotated boundary, that the bulk dipolar Chern-Simons theory is invariant under the parity of $y$.

## 3.2 Triangular dipole anomaly

Consider a two-dimensional ($x$-$y$ plane) triangular symmetry [54, 55]. There exists a dipolar Chern-Simons term

$$S_{CS} = C_\triangle \int d^4x\ \epsilon^{\mu\nu\rho\sigma} A_\mu^a \partial_\nu A_\rho^b \delta_\sigma^c f_{abc}^\triangle, \tag{28}$$

with $f_{abc}^\triangle = \delta_{ax}\sigma_{bc}^x + \delta_{ay}\sigma_{bc}^z$, where $\sigma^{x,y,z}$ are three Pauli matrices. There is another invariant tensor $f_{abc}^{\triangle\prime} = \delta_{ax}\sigma_{bc}^z - \delta_{ay}\sigma_{bc}^x$ corresponding to rotating the triangle by 180 degrees, and the two invariant tensors are related by $\epsilon_{da}f_{abc}^{\triangle\prime} = f_{dbc}^\triangle$. This dipolar Chern-Simons term breaks time-reversal symmetry and parity of $x$ and $z$ but preserves the parity of $y$ due to the oddity of $f_{abc}^\triangle$ under $y \to -y$.

Let us first consider putting a boundary at $y = 0$. Under the gauge transformation, (28) changes by

$$\delta S_{CS}(y = 0) = C_\triangle \int dt dx dy dz\ \epsilon^{\mu\nu\rho\sigma} \partial_\mu(\xi^b \partial_\nu A_\rho^c \delta_\sigma^d f_{bcd}^\triangle) = C_\triangle \int dt dx dz\ \epsilon^{\nu\rho x} \xi^a \partial_\nu A_\rho^b f_{abx}^\triangle. \tag{29}$$

The corresponding chiral boson action is given by

$$S[\phi, \phi^a] = \int dt dx dz\ \frac{\chi}{2}(\partial_t\phi)^2 - \frac{K_n}{2}(\partial_I\phi + \phi_I)^2 - C_\triangle \partial_t\phi^a \partial_z\phi^b f_{abx}^\triangle + \frac{\kappa}{2}(\partial_t\phi^a)^2, \tag{30}$$

where $I = x, z$. Varying the action and using $\phi_I \approx -\partial_I\phi$, we arrive at the equations of motion

$$\chi \partial_t\phi - 2C_\triangle \partial_x\partial_z\phi_y = 0,$$
$$\kappa \partial_t\phi_y + 2C_\triangle \partial_x\partial_z\phi = 0. \tag{31}$$

Upon Fourier transformation, it leads to two modes

$$\omega = \pm\sqrt{\frac{4C_\triangle^2}{\kappa\chi}|k_x k_z|}. \tag{32}$$

Similar to the tetrahedral dipole anomaly, the counterpropagating modes here are quadratic.

If the boundary is at $x = 0$, the action changes by

$$\delta S_{CS}(x=0) = C_\triangle \int dtdxdydz\ \epsilon^{\mu\nu\rho\sigma}\partial_\mu(\xi^b\partial_\nu A_\rho^c\delta_\sigma^d f_{bcd}^\triangle) = C_\triangle \int dtdydz\ \epsilon^{\nu\rho y}\xi^a\partial_\nu A_\rho^b f_{aby}^\triangle. \tag{33}$$

The corresponding chiral action is given by

$$S[\phi,\phi^a] = \int dtdydz\ \frac{\chi}{2}(\partial_t\phi)^2 - \frac{K_n}{2}(\partial_I\phi + \phi_I)^2 - C_\triangle\partial_t\phi^a\partial_z\phi^b f_{aby}^\triangle + \frac{\kappa}{2}(\partial_t\phi^a)^2. \tag{34}$$

Varying the action and using $\phi_I \approx -\partial_I\phi$, we arrive at the equations of motion

$$\chi\partial_t\phi - 2C_\triangle\partial_y^2\partial_z\phi = 0\,,$$
$$\kappa\partial_t\phi_x - 2C_\triangle\partial_z\phi_x = 0\,. \tag{35}$$

Upon Fourier transformation, it leads to two chiral modes

$$\omega = \frac{2C_\triangle}{\chi}k_y^2 k_z\,, \tag{36}$$

$$\omega = -\frac{2C_\triangle}{\kappa}k_z\,, \tag{37}$$

with both linear and cubic dispersion relations. As promised by the bulk dipolar Chern-Simons theory, the chiral modes are odd under parity of $z$ but even under parity of $y$.

We can further consider an arbitrary boundary perpendicular to the $x$-$y$ plane. Instead of rotating the coordinate, we rotate the invariant tensor through $f_{acd}^{\triangle,R} = R_{ab}f_{bcd}^\triangle$. Parametrizing the rotation matrix by $\theta$, we have

$$f_{acd}^{\triangle,R} = \delta_{a,x}(\cos\theta\sigma_{cd}^x - \sin\theta\sigma_{cd}^z) + \delta_{a,y}(\sin\theta\sigma_{cd}^x + \cos\theta\sigma_{cd}^z) = \cos\theta f_{acd}^\triangle - \sin\theta f_{acd}^{\triangle\prime}. \tag{38}$$

Let the boundary be at $y = 0$, so the action changes by

$$\delta S_{CS}(y=0) = C_\triangle \int dtdxdydz\ \epsilon^{\mu\nu\rho\sigma}\partial_\mu(\xi^b\partial_\nu A_\rho^c\delta_\sigma^d f_{bcd}^{\triangle,R}) = C_\triangle \int dtdxdz\ \epsilon^{\nu\rho x}\xi^a\partial_\nu A_\rho^b f_{abx}^{\triangle,R}. \tag{39}$$

The corresponding chiral boson action is given by

$$S[\phi,\phi^a] = \int dtdxdz\ \frac{\chi}{2}(\partial_t\phi)^2 - \frac{K_n}{2}(\partial_I\phi + \phi_I)^2 - C_\triangle\partial_t\phi^a\partial_z\phi^b f_{abx}^{\triangle,R} + \frac{\kappa}{2}(\partial_t\phi^a)^2. \tag{40}$$

Varying the action and using $\phi_I \approx -\partial_I\phi$, we arrive at the equations of motion

$$\chi\partial_t\phi - 2C_\triangle\sin\theta\partial_x^2\partial_z\phi - 2C_\triangle\cos\theta\partial_x\partial_z\phi_y = 0\,,$$
$$\kappa\partial_t\phi_y - 2C_\triangle\sin\theta\partial_z\phi_y + 2C_\triangle\cos\theta\partial_x\partial_z\phi = 0\,. \tag{41}$$

When $\theta = 0$, we get back (32). However, when $\theta \neq 0$, it will lead to two chiral modes (to the leading order in wavevector)

$$\omega \approx \frac{2C_\triangle}{\chi\sin\theta}k_x^2 k_z\,, \tag{42}$$

$$\omega \approx -\frac{2C_\triangle\sin\theta}{\kappa}k_z\,. \tag{43}$$

It is consistent with (36) when setting $\theta = \pi/2$.

If the boundary is at $z = 0$, the action changes by

$$\delta S_{CS}(z=0) = C_\triangle \int \mathrm{d}t\mathrm{d}x\mathrm{d}y\mathrm{d}z \; \epsilon^{\mu\nu\rho\sigma} \partial_\mu(\xi^b \partial_\nu A_\rho^c \delta_\sigma^d f_{bcd}^\triangle) = C_\triangle \int \mathrm{d}t\mathrm{d}x\mathrm{d}y \; \epsilon^{\nu\rho I} \xi^a \partial_\nu A_\rho^b f_{abI}^\triangle.$$
(44)

The corresponding chiral boson action is given by

$$S[\phi, \phi^a] = \int \mathrm{d}t\mathrm{d}y\mathrm{d}z \; \frac{\chi}{2}(\partial_t \phi)^2 - \frac{K_n}{2}(\partial_I \phi + \phi_I)^2 - C_\triangle \epsilon^{IJ} \partial_t \phi^a \partial_I \phi^b f_{abJ}^\triangle + \frac{\kappa}{2}(\partial_t \phi^a)^2. \quad (45)$$

Varying the action and using $\phi_I \approx -\partial_I \phi$, we arrive at the equations of motion

$$\chi \partial_t \phi + 2C_\triangle(\partial_x^3 - 3\partial_y^2 \partial_x)\phi = 0. \qquad (46)$$

Upon Fourier transformation, it leads to a single chiral mode

$$\omega = -\frac{2C_\triangle}{\chi}(k_x^3 - 3k_y^2 k_x), \qquad (47)$$

with cubic dispersion relation. As promised by the bulk dipolar Chern-Simons theory, the chiral mode is odd under parity of $x$ but even under parity of $y$. The dipolar Chern-Simons term does not involve the $z$-dipole that is out of the plane, so there is no linear-dispersing mode.

## 3.3 Boundary-dependent bulk-edge correspondence

When the boundary is orthogonal to the direction along which the parity is preserved by the bulk dipolar Chern-Simons theory, the boundary gapless modes will form a time-reversal pair. These modes have a different universal behavior from the chiral modes at other boundaries. For instance, they are quadratically dispersing and counterpropagating against each other.[5] In fact, unlike the chiral modes, those non-chiral modes are protected by additional symmetries. They are protected by the parity normal to the boundary. Since the bulk is even under the parity normal to the boundary, the out-of-plane gapless mode is forbidden in the equation of motion. Hence, the out-of-plane dipole has to couple to the charge, which is bonded to the in-plane dipole, to form a paired mode. We further justified it by considering symmetry-breaking perturbations. This is amount to choosing different boundaries. We have seen that as soon as the parity normal to the boundary is not a symmetry of the bulk, the non-chiral modes will become two chiral modes with different dispersion at the leading order in wavevectors. In a word, the non-chiral modes at specific boundaries correspond to some symmetry-protected topological (SPT) bulk, while the chiral modes at generic boundaries correspond to a more general bulk that does not require additional symmetries, like quantum Hall state for example. Moreover, observe that such non-chiral modes cannot happen in a single-specie fluid, like the $U(1)$ chiral anomaly, and is instead carried by a mixture of the dipole moments that are in the plane and out of the plane. Therefore, we suggest calling it the *mixed anomaly* between the in-plane dipole moment and out-of-plane dipole moment. As a result, we find that the dipole anomaly with point group symmetries would *violate* the conventional bulk-edge correspondence: A single bulk dipolar Chern-Simons theory may lead to different boundary anomalies at different boundaries,[6] and, at the same time, these different boundary anomalies would correspond to either SPT or a generic bulk depending on whether there is an additional symmetry that is protecting the boundary modes.

---

[5]This type of boundary anomalous flow is similar to the quantum spin Hall edge state, but the symmetries that protect them are different.

[6]Similar observations have been made in a recent work about subsystem anomaly [56]. In the meantime, it was noticed that a single boundary anomaly may correspond to different bulk fracton models [57–59].

Following the analysis at the end of Section 2 and assuming isotropy, the zeroth-order dissipative fixed point for the *cubic* chiral modes in $d = 2$ would be $z \approx 4$. This is equal to the subdiffusive scaling at the linear level, so $d = 2$ is the critical dimension. On the other hand, the zeroth-order dissipative fixed point for the *quadratic* modes in $d = 2$ would be $z \approx 3$. This is still below $z = 4$ from linear analysis, so we anticipate a new dissipative fixed point. In a word, a single bulk dipolar Chern-Simons theory could support different dissipative fixed points on different boundaries.

As a final remark, the gapless edge modes would lead to UV/IR mixing [26]. For example, taking $k_x - k_y = 0$ in (23), the low energy state $\omega = 0$ talks to both small wavevector $k_x + k_y \ll 1$ and large wavevector $k_x + k_y \gg 1$. This type of dispersion relation is also relevant for the exciton Bose liquid [60,61].

## 4 Analogy to torsional anomaly in $U(1)$ quantum Hall state

In this section, we take a detour to consider a $U(1)$ quantum Hall state in $D = 2 + 1$ *without* dipole symmetry. However, there is an emergent nonlinear dipole-like symmetry on the Lowest Landau level (LLL) as recently discussed in [62]. This is realized as the volume-preserving diffeomorphism (VPD). Consider a spatial Lie derivative

$$\mathcal{L}_{\chi^i} e^a_\mu = \chi^i \partial_i e^a_\mu + \partial_\mu \chi^i \delta^a_i \,, \tag{48}$$

and $\partial_i \chi^i = 0$ for VPD. Notice that the linear term $\partial_\mu \chi^i \delta^a_i$ looks just like the dipole gauge transformation in (1). By coupling to the background $U(1)$ gauge field $A_\mu$ and fixing $A_i = -\frac{1}{2} B \epsilon_{ij} x_j$, we arrive at the equations of motion (in the flat spacetime limit) [62]

$$\begin{aligned}\partial_\mu J^\mu &= 0 \,, \\ \partial_i T^i_a &= B \epsilon_{ab} J^i \delta_{ib} \,. \end{aligned} \tag{49}$$

This coincides with the hydrodynamic equations under a magnetic field and in the LLL limit by letting $m \to 0$. Let us now introduce a Chern-Simons term in $D = 3 + 1$ mimicking (18):

$$S_{\text{torsion}} = C' \int \mathrm{d}^4 x \; \epsilon^{\mu\nu\rho\sigma} e^a_\mu \partial_\nu e^b_\rho \delta^c_\sigma f_{abc} \,, \tag{50}$$

and we find that this is the generalized torsional Chern-Simons theory [63,64]. Under the nonlinear diffeomorphism (48), (50) changes by

$$\begin{aligned}\delta S_{\text{torsion}} &= C' \int \mathrm{d}^4 x \; \epsilon^{\mu\nu\rho\sigma} \left( \chi^i \partial_i e^a_\mu \partial_\nu e^b_\rho + \partial_\mu \chi^i \delta^a_i \partial_\nu e^b_\rho + e^a_\mu \partial_\nu (\chi^i \partial_i e^b_\rho + \partial_\rho \chi^i \delta^b_i) \right) \delta^c_\sigma f_{abc} \\ &= C' \int \mathrm{d}^4 x \; \epsilon^{\mu\nu\rho\sigma} \left( \chi^i \partial_i e^a_\mu G^b_{\nu\rho} + \partial_\mu \chi^i \delta^a_i G^b_{\nu\rho} \right) \delta^c_\sigma f_{abc} \\ &= -C' \int \mathrm{d}^4 x \; \epsilon^{\mu\nu\rho\sigma} \delta^a_\mu \partial_i \chi^i G^b_{\nu\rho} \delta^c_\sigma f_{abc} = 0 \,, \end{aligned} \tag{51}$$

where we approximated the spacetime to have constant torsion $G^a_{\mu\nu} = \partial_\mu e^a_\nu - \partial_\nu e^a_\mu$ and to be close to the flat limit, and in the last step we used VPD. To obtain the boundary anomaly, let us consider a triangular symmetry in the $x$-$y$ plane, and place the boundary at $z = 0$. The

boundary terms from (51) are given by

$$
\begin{aligned}
\delta S_{\text{torsion}} &= C' \int d^3x \; \epsilon^{\mu z \rho c} e^a_\mu (\chi^i \partial_i e^b_\rho + \partial_\rho \chi^i \delta^b_i) f_{abc} + \epsilon^{z \nu \rho c} \chi^i \delta^a_i G^b_{\nu\rho} f_{abc} \\
&= C' \int d^3x \; \epsilon^{z \nu \rho c} \chi^i \delta^a_i G^b_{\nu\rho} f_{abc} \,.
\end{aligned}
\tag{52}
$$

The first term vanishes because $f_{abc}$ is fully symmetric. Then, the torsional Chern-Simons term changes the equations of motion in (49) to

$$
\begin{aligned}
\partial_\mu J^\mu &= 0 \,, \\
\partial_i T^i_a &= B \epsilon_{ab} J^i \delta_{ib} + C' \epsilon^{\mu \nu c} G^b_{\mu\nu} f_{abc} \,.
\end{aligned}
\tag{53}
$$

Consequently, rather than being subdiffusive [65], the $U(1)$ quantum Hall state under torsional anomaly will develop a cubic dispersing chiral mode just like in (47).

# 5 Generalized particle-vortex duality with dipole symmetry

Particle-vortex duality typically maps between particle excitations in one theory and vortex excitations in another theory. The conventional particle-vortex duality describes a mapping between the following two models:

$$
\text{XY model}: \quad S = \int d^3x |(\partial_\mu - iA_\mu)\Phi|^2 + V(\Phi) \,,
\tag{54a}
$$

$$
\text{Higgs model}: \quad S = \int d^3x |(\partial_\mu - ia_\mu)\tilde{\Phi}|^2 + \tilde{V}(\tilde{\Phi}) + \frac{1}{2\pi} \epsilon^{\mu\nu\rho} A_\mu \partial_\nu a_\rho \,,
\tag{54b}
$$

where $A_\mu$ is the background gauge field. The essence of the particle-vortex duality is realized through the mixed Chern-Simons term involving the dynamical gauge field $a_\mu$, which reveals that the charge density associated with $\Phi$ is equal to the flux density $da/2\pi$. Therefore, the Goldstone mode of spontaneously symmetry broken XY model is identified with the dual photon in the Higgs model. The exact same idea can be generalized to spontaneously broken dipole symmetry: The dipole symmetry acts just like a $U(1)$ symmetry for dipole moments. Imagine dipole shifts on the dipole scalar field $\Phi^a \to \Phi^a e^{i\xi^a}$, where $a = 1, 2, \dots$ are just labels. Then, we can immediately write down the particle-vortex duality for dipoles:

$$
\text{dipole}-\text{XY model}: \quad S = \sum_a \int d^3x |\partial_\mu \Phi^a - iA^a_\mu \Phi^a|^2 + V(\Phi^a) \,,
\tag{55a}
$$

$$
\text{dipole}-\text{Higgs model}: \quad S = \sum_a \int d^3x |\partial_\mu \tilde{\Phi}^a - ia^a_\mu \tilde{\Phi}^a|^2 + \tilde{V}(\tilde{\Phi}^a) + \frac{1}{2\pi} \epsilon^{\mu\nu\rho} A^a_\mu \partial_\nu a^a_\rho \,.
\tag{55b}
$$

Note there are no $U(1)$ degrees of freedom in these dipole models. A more interesting scenario is to include $U(1)$ symmetry. However, since we can not have the $U(1)$ Chern-Simons term in the presence of dipole symmetry, including $U(1)$ symmetry will not bring about additional duality in terms of the flux attachment. To have a non-trivial duality, we need to condense the dipole first, and then study the remaining $U(1)$ phase transition; such $U(1)$ symmetry breaking has been studied in [42], and the condensed phase was argued to be Lifshitz-like.

Let us call the dipole Goldstone $\varphi^a$, and it transforms under the dipole shift as

$$
\varphi^a \to \varphi^a + \xi^a \,.
\tag{56}
$$

It is useful to define the modified $U(1)$ gauge fields

$$B_\mu[\varphi_a] \equiv \hat{A}_\mu - x_a A_\mu^a - \delta_{\mu a}\varphi^a, \tag{57a}$$

$$b_\mu[\varphi_a] \equiv \hat{a}_\mu - x_a A_\mu^a - \delta_{\mu a}\varphi^a, \tag{57b}$$

where $\hat{A}_\mu, \hat{a}_\mu$ transform as normal $U(1)$ gauge field like in Section 2. We propose a new type of particle-vortex duality mediated by dipole Goldstone:

$$\varphi^a-\text{XY model}: \quad S = \int d^3x |(\partial_\mu - iB_\mu[\varphi^a])\Phi|^2 + (\partial_\mu\varphi^a - A_\mu^a)^2 + V(\Phi), \tag{58a}$$

$$\tilde{\varphi}^a-\text{Higgs model}: \quad S = \int d^3x |(\partial_\mu - ib_\mu[\tilde{\varphi}^a])\tilde{\Phi}|^2 + (\partial_\mu\tilde{\varphi}^a - A_\mu^a)^2 + \tilde{V}(\tilde{\Phi})$$

$$+ \frac{1}{2\pi}\epsilon^{\mu\nu\rho}B_\mu[\tilde{\varphi}^a]\partial_\nu b_\rho[\tilde{\varphi}^a]. \tag{58b}$$

Both models have a global dipole symmetry: The shift $\Phi \to \Phi e^{-ix^a\xi^a}$ is canceled by the shift of $-x_a A_\mu^a - \delta_{\mu a}\varphi^a$ in (57), and the Chern-Simons term in (58b) is invariant up to a total derivative. Like (54), $\varphi^a$–XY model has a $U(1)$ global symmetry, while $\tilde{\varphi}^a$–Higgs model has a $U(1)$ gauge symmetry.

There are two phases in $\varphi^a$–XY model: $i$) unbroken $U(1)$, and $ii$) spontaneously broken $U(1)$. Case $i$ has gapped $\Phi$ excitations but linear-dispersing gapless $\varphi^a$. In case $ii$, we can parametrize the scalar field as $\Phi = \rho e^{i\phi}$, and neglect the massive excitations of $\rho$. The low-energy theory becomes

$$S_{U(1)\text{ broken}} = \int d^3x \, (\partial_\mu\phi + \delta_{\mu a}\varphi^a)^2 + (\partial_\mu\varphi^a)^2 \to \int d^3x \, (\partial_t\phi)^2 + (\partial_i\partial_a\phi)^2, \tag{59}$$

where we used the fact that the dipole Goldstone is gapped by the $U(1)$ Goldstone: $\partial_\mu\phi = -\delta_{\mu a}\varphi^a$. The low-energy excitations are $\omega \sim k^2$ and thus is Lifshitz-like. In this phase, the theory also has vortex excitations. Unlike the usual superfluid vortex, the $U(1)$ vortex here is given by [8]

$$-\oint dx^i \, x^a \partial_i\partial_a\phi = -\oint dx^i \, \partial_i(x^a\partial_a\phi) + \oint dx^i \partial_i\phi = 2\pi n, \quad n \in \mathbb{Z}. \tag{60}$$

On dimensional ground, such a vortex is gapped since it is logarithmically confined.

Let us now look at the $\tilde{\varphi}^a$–Higgs model, which also has two phases: $i$) unbroken $U(1)$ gauge symmetry, and $ii$) broken $U(1)$ gauge symmetry. In case $i$, the dynamical gauge field $b_\mu$ will support dual photon $\sigma$ excitations. To see it, we ignore the coupling to the field $\tilde{\Phi}$ and introduce the Maxwell term. The partition function reads

$$Z = \int Db \exp\left(i\int d^3x - \frac{1}{4g^2}f^2 - \frac{1}{4\pi}\epsilon^{\mu\nu\rho}\delta_{\mu a}\tilde{\varphi}^a f_{\nu\rho}\right)$$

$$= \int Df D\sigma \exp\left(i\int d^3x - \frac{1}{4g^2}f^2 - \frac{1}{4\pi}\epsilon^{\mu\nu\rho}\delta_{\mu a}\tilde{\varphi}^a f_{\nu\rho} + \frac{1}{4\pi}\epsilon^{\mu\nu\rho}\sigma\partial_\mu f_{\nu\rho}\right), \tag{61}$$

where $f_{\mu\nu} = \partial_\mu b_\nu - \partial_\nu b_\mu$, and $\sigma$ is a Lagrangian multiplier implementing the Bianchi identity $\epsilon^{\mu\nu\rho}\partial_\mu f_{\nu\rho} = 0$. Using the equation of motion

$$f^{\mu\nu} = -\frac{g^2}{2\pi}\epsilon^{\mu\nu\rho}\left(\partial_\mu\sigma + \delta_{\mu a}\tilde{\varphi}^a\right), \tag{62}$$

we can integrate out $f_{\mu\nu}$ to obtain an effective action

$$S_{\text{dual photon}} = \int d^3x \, \frac{g^2}{8\pi^2}(\partial_\mu\sigma + \delta_{\mu a}\tilde{\varphi}^a)^2. \tag{63}$$

Compared to (59), we identify $\sigma$ with $\phi$ as the hallmark of the particle-vortex duality. Adding back the dipole Goldstone dynamics, we get the Lifshitz scaling at low energy. In the meantime, the $\tilde{\Phi}$ excitations are gapped and their interactions are mediated by the Coulomb interaction. In $d = 2$, Coulomb energy has logarithmic divergence, so it matches the vortex excitations in the $\varphi^a$−XY model. Therefore, case $i$ in $\tilde{\varphi}^a$−Higgs model is dual to case $ii$ in $\varphi^a$−XY model.

One may wonder if the dipole Goldstone in the dual theory is still "gapped" when the dynamical gauge field is turned off. Naively, this would lead to a linear-dispersing dipole Goldstone that is in tension with the Lifshitz theory. Thanks to the Chern-Simons term in (58b), the correct low-energy theory in the dual picture is given by

$$S_{\text{dual}} = \int d^3x \; -\frac{1}{2\pi}\epsilon_{ab}\tilde{\varphi}^a\partial_t\tilde{\varphi}^b + (\partial_i\tilde{\varphi}^a)^2, \tag{64}$$

where we neglected the second-order time derivatives. This theory has a Lifshitz fixed point $\omega \sim k^2$. The first-order time derivative term indicates that $\tilde{\varphi}^x$ and $\tilde{\varphi}^y$ are canonically conjugate to each other reminiscent of the Heisenberg ferromagnet [66]. In fact, the analogy is not a coincidence – dipole moment of vorticity (recall the dual theory is a theory of vortex) does not commute with itself: [67]

$$\{D_i, D_j\} = -\epsilon_{ij}\Gamma, \tag{65}$$

where $\{,\}$ is the Poisson bracket, and $\Gamma$ is the total vortex. Think of $\Gamma$ as the total spin, the dipole moment has an identical algebra as the Heisenberg ferromagnet, and importantly, it allows for the first-order time derivative term to appear in (64). Thus the two linear-dispersing Goldstone will couple together to form a single quadratic-dispersing Goldstone. To summarise, the Chern-Simons term in (58b) plays two roles in the dual theory. First, it generates the usual flux attachment to relate the gauge vortex $\sigma$ to the real particle $\Phi$. Second, it reflects the non-commutativity of dipole moments in the dual picture and leads to a Lifshitz fixed point.

Lastly, in case $ii$, $\tilde{\Phi}$ acquires an expectation value, so the $U(1)$ gauge symmetry is broken, and the dual photon $\sigma$ becomes massive. This results in $b_\mu = 0$, so the dipole Goldstone becomes linear-dispersing. This phase is thus dual to case $i$ in $\varphi^a$−XY model.

## 6 Dipolar Chern-Simons theory on curved spacetime

In $D = 2 + 1$, the usual $U(1)$ Chern-Simons theory can be written in the differential form $\int A \wedge F$, where $F = dA$ is the two-form field strength. In this way, the $U(1)$ Chern-Simons term is invariant under $U(1)$ gauge transformation in a generic closed manifold, and the integral does not depend on the metric meaning it is a topological field theory. However, a naive generalization to dipolar Chern-Simons theory like $\int A^a \wedge F^b f_{ab}$ with $F^b = dA^b$ is not correct because this term is not invariant under the dipole gauge transformation on curved spacetime [19]

$$A_\mu^a \to A_\mu^a + \nabla_\mu\xi^a, \tag{66}$$

where $\nabla_\mu\xi^a = \partial_\mu\xi^a + \omega_{\mu b}^a\xi^b$, and $\omega_{\mu b}^a$ is the spin connection. It turns out that it is impossible to find a dipole field strength that is invariant under (66) since any vector charge gauge field will necessarily encounter the Ricci curvature when computing its own "curvature". The reason is that the dipole algebra is "non-abelian" in spacetime:

$$[P_b, D_c] = -iQ\delta_{bc}, \tag{67a}$$

$$[P_d, L_{bc}] = i(\delta_{dc}P_b - \delta_{db}P_c), \tag{67b}$$

$$[D_d, L_{bc}] = i(\delta_{dc}D_b - \delta_{db}D_c), \tag{67c}$$

$$[L_{bc}, L_{de}] = i(\delta_{bd}L_{ce} - \delta_{be}L_{cd} - \delta_{cd}L_{be} + \delta_{ce}L_{bd}), \tag{67d}$$

where $P_a$ generates translation symmetry, $D_a$ generates dipole symmetry, and $L_{ab}$ generates $SO(d)$ rotational symmetry.

To build a gauge invariant theory on curved spacetime, we need to either cancel (66) by coupling to matter fields or treat the dipole symmetry as a part of the spacetime symmetry in (67). The former is presented in Section 6.1 to include dipole Goldstones in the dipolar Chern-Simons theory. The latter leads to a non-abelian Chern-Simons theory in Section 6.2.

Before constructing the field theories, we review some necessary ingredients of the vielbein formalism; for a complete analysis of dipole symmetry on curved spacetime, we refer readers to [19]. The covariant derivative is defined as

$$\nabla_\mu e_\nu^0 = \partial_\mu e_\nu^0 - \Gamma_{\mu\nu}^\rho e_\rho^0 \,, \tag{68a}$$

$$\nabla_\mu e_\nu^b = \partial_\mu e_\nu^b + \omega_{\mu c}^b e_\nu^c - \Gamma_{\mu\nu}^\rho e_\rho^b \,, \tag{68b}$$

where $\Gamma_{\mu\nu}^\rho$ is the Christoffel connection, and $\omega_{\mu b}^a$ is the spin connection for space only. Since the spin connection and the Christoffel connection are not independent, and it is more natural to treat the spin connection as the gauge field for spatial rotational symmetry (see (79)), we impose the metric compatibility condition

$$\nabla_\mu e_\nu^a = 0 \,, \tag{69}$$

such that the Christoffel connection can be expressed in terms of the spin connections and the vielbeins. We will further fix the temporal component of the vielbein to be $e_\mu^0 = \delta_\mu^0$ since the time translation generator is central in the dipole algebra (67) so its gauge field $e_\mu^0$ is decoupled from the theory we are interested in. Now, the vielbein is full rank, and its inverse is defined through

$$e_\mu^\alpha e_\beta^\mu = \delta_\beta^\alpha \,, \qquad e_\mu^\alpha e_\alpha^\nu = \delta_\mu^\nu \,. \tag{70}$$

## 6.1 Couple to dipole Goldstone

Define the dipole field strength as [19]

$$F_{\mu\nu}^a \equiv \partial_\mu A_\nu^a - \partial_\nu A_\mu^a + \omega_{\mu b}^a A_\nu^b - \omega_{\nu b}^a A_\mu^b \,. \tag{71}$$

It transforms under dipole shift (66) as $F_{\mu\nu}^a \to F_{\mu\nu}^a + R_{b\mu\nu}^a \xi^b$, with

$$R_{c\mu\nu}^b \equiv \partial_\mu \omega_{\nu c}^b - \partial_\nu \omega_{\mu c}^b + \omega_{\mu d}^b \omega_{\nu c}^d - \omega_{\nu d}^b \omega_{\mu c}^d \,, \tag{72}$$

the curvature tensor. In order to have a gauge-invariant field strength, we introduce the *dipole Goldstone* $\varphi^b$ that transforms under dipole shift as

$$\varphi^a \to \varphi^a + \xi^a \,. \tag{73}$$

The gauge-invariant field strength is defined as

$$\tilde{F}_{\mu\nu}^a \equiv F_{\mu\nu}^a - R_{b\mu\nu}^a \varphi^b \,. \tag{74}$$

Now, $\int A^a \wedge \tilde{F}^b f_{ab}$ is still not gauge-invariant, but transforms as

$$\delta \int A^a \wedge \tilde{F}^b f_{ab} = \int \mathrm{d}^3 x \, \partial_\mu \left( e \epsilon^{\mu\nu\rho} \xi^a \tilde{F}_{\nu\rho}^b f_{ab} \right) - \int \mathrm{d}^3 x \, e \xi^a \nabla_\mu' \left( \epsilon^{\mu\nu\rho} \tilde{F}_{\nu\rho}^b \right) f_{ab} \,, \tag{75}$$

where $\nabla'_\mu \equiv \nabla_\mu + 2\Gamma^\nu_{[\mu\nu]}$, and we used $\nabla_\mu f_{ab} = 0$. While the first term is a boundary term, the second term needs an additional term $\varphi^a \nabla'_\mu \left( \epsilon^{\mu\nu\rho} \tilde{F}^b_{\nu\rho} \right)$ to cancel it. Collecting the above results, we arrive at the final gauge-invariant dipolar Chern-Simons term:

$$S_{CS} = C \int \mathrm{d}^3 x \, e \left[ \epsilon^{\mu\nu\rho} A^a_\mu \tilde{F}^b_{\nu\rho} + \varphi^a \nabla'_\mu \left( \epsilon^{\mu\nu\rho} \tilde{F}^b_{\nu\rho} \right) \right] f_{ab} \,. \tag{76}$$

The inclusion of the dipole Goldstone makes physical sense – when dipoles want to form cyclotron motions in the quantum Hall phase on the curved spacetime, due to the kinetic constraint, the dipoles must be generated out of the condensate.

We expect that (76) is relevant to the construction of parity-violating dipole hydrodynamics generalizing [19]. This is a useful starting point since there is no known dipolar anomaly on curved spacetime to our knowledge. One can then follow the analysis of [68] in a reverse way to determine various parity-odd transport coefficients in terms of the coefficient $C$ [69].

## 6.2 Non-abelian Chern-Simons theory of 3D gravity: Fracton-elasticity duality and beyond

We follow the non-relativistic construction in [43, 44] to build a non-abelian Chern-Simons theory based on (67); see also the original proposal of relativistic 3D Chern-Simons gravity in [45, 46].[7] A non-abelian Chern-Simons theory is given by[8]

$$S_{CS} = C \int_{M^3} \mathrm{tr} \left( \mathcal{A} \wedge \mathrm{d}\mathcal{A} + \mathrm{i}\frac{2}{3} \mathcal{A} \wedge \mathcal{A} \wedge \mathcal{A} \right) \,, \tag{77}$$

where the trace encodes a non-degenerate invariant bilinear form on the dipole algebra. The crucial difference from the Galilean algebra in [43, 44] is that we are allowed to turn off the time-translation generator $H$, which is central in the dipole algebra, and the invariant bilinear form is already non-degenerate without further central extensions. This can be seen by observing that the bilinear form $\epsilon^{ab} D_a P_b - \frac{1}{2} Q \epsilon^{ab} L_{ab}$ is invariant and commutes with all the generators. Hence, we are interested in the following non-degenerate bilinear form

$$\langle D_a, P_b \rangle = \epsilon_{ab} \,, \qquad \langle Q, L_{ab} \rangle = -\frac{1}{2} \epsilon_{ab} \,. \tag{78}$$

The gauge field $\mathcal{A}$ is locally given by a dipole-algebra-valued one-form

$$\mathcal{A} = e^a P_a + \frac{1}{2} \omega^{ab} L_{ab} + A^a D_a + A Q \,, \tag{79}$$

where we can identify $e^a_\mu$ as the vielbein, $\omega^{ab}_\mu$ as the spin connection, $A^a_\mu$ as the dipole gauge field and $A_\mu$ as the $U(1)$ gauge field. First, we have

$$-\mathrm{i}\mathcal{A} \wedge \mathcal{A} = e^a \wedge \omega^{ba} P_b - e^a \wedge A^a Q + A^a \wedge \omega^{ba} D_b + \omega^{ac} \wedge \omega^{bc} L_{ab} \,. \tag{80}$$

The two-form curvature is then defined as

$$\mathcal{F} \equiv \mathrm{d}\mathcal{A} + \mathrm{i}\mathcal{A} \wedge \mathcal{A} = R^a(P) P_a + R^a(D) D_a + R^{ab}(L) L_{ab} + R(Q) Q \,, \tag{81}$$

---

[7]After this work was completed and posted, we learned that the Carrollian gravity [70] has a similar structure as our dipole gravity due to their isomorphic algebra [36]. We anticipate that the Carrollian gravity is relevant to gauging the spacetime dipole symmetry proposed by [71] (see [72]).

[8]The level $C$ is in general not quantized due to noncompact spacetime symmetries, but its precise value will not be important in our discussions.

where the field strengths are given by

$$R^a(P) = \mathrm{d}e^a + \omega^{ab} \wedge e^b \,, \tag{82a}$$

$$R^a(D) = \mathrm{d}A^a + \omega^{ab} \wedge A^b \,, \tag{82b}$$

$$R^{ab}(L) = \mathrm{d}\omega^{ab} + \omega^{ac} \wedge \omega^{cb} \,, \tag{82c}$$

$$R(Q) = \mathrm{d}A + e^a \wedge A^a \,. \tag{82d}$$

The dipole field strength $R^a(D)$ and the Ricci curvature tensor $R^{ab}(L)$ agree with (71) and (72), respectively. Note that in $D = 2+1$, we can write

$$\omega^{ab} \equiv \omega \epsilon^{ab} \,, \tag{83}$$

so $\omega^{ac} \wedge \omega^{bc} = 0$ is trivial. Now, consider a four-dimensional manifold $M^4$. A topological invariant is given by, according to (78),

$$\int_{M^4} \mathrm{tr}(\mathcal{F} \wedge \mathcal{F}) = \int_{M^4} \epsilon_{ab} \left[ (\mathrm{d}A^a + \omega^{ac} \wedge A^c) \wedge (\mathrm{d}e^b + \omega^{bc} \wedge e^c) - \frac{1}{2}(\mathrm{d}A + e^c \wedge A^c) \wedge \mathrm{d}\omega^{ab} \right]$$

$$= \int_{M^4} \mathrm{d}\left[ \epsilon_{ab} A^a \wedge \mathrm{d}e^b - A \wedge \mathrm{d}\omega - A^a \wedge \omega \wedge e^a \right]. \tag{84}$$

Because of the closed form, the integral can be reduced to its 3D boundary $M^3 \equiv \partial M^4$, which is by definition the Chern-Simons action (77):

$$S_{CS} = C \int_{M^3} \epsilon_{ab} R^a(D) \wedge e^b - A \wedge \mathrm{d}\omega = C \int_{M^3} \epsilon_{ab} A^a \wedge R^b(P) - A \wedge \mathrm{d}\omega \,. \tag{85}$$

From the construction, (85) is automatically invariant under the non-abelian gauge transformation $\delta \mathcal{A} = \mathrm{d}\Lambda + i\mathcal{A} \wedge \Lambda$. As a remark, the gauge transformation in (1) is precisely the flat spacetime limit of this non-abelian gauge transformation. Interestingly, the second term itself in (85) is known as the $U(1)$ Wen-Zee term [47], thus, for reasons we shall shortly explain, we call (85) the *dipolar Wen-Zee term*. Before a detailed linear response analysis, several remarks follow. The term $\omega \wedge \mathrm{d}\omega$ (known as the second Wen-Zee term) can be generated by allowing a bilinear form $\langle J_{ab}, J_{cd} \rangle = \epsilon_{ab}\epsilon_{cd}$ [44], but we neglect it as it decouples from the dipole symmetry. We also neglected the gravitational Chern-Simons term due to framing anomaly [73]. There is no dipolar Chern-Simons term $A^a \wedge \mathrm{d}A^a$ because the bilinear form $\delta^{ab} D_a D_b$ does not commute with $P_a$ (see Section 6.1 for dipolar Chern-Simons theory on curved spacetime).

Let us first review the conventional Wen-Zee term. A linear response analysis gives a shift to the charge density $\delta S/\delta A_0 = -C\epsilon^{ij}\partial_i \omega_j$ and a spin current $\delta S/\delta \omega_\mu = -C\epsilon^{\mu\nu\rho}\partial_\nu A_\rho$. Knowing the vortex in solids corresponds to defects of rotational symmetry described by curvature or disclination [74], we find a correspondence that each charge/boson is attached by a flux of the spin connection, i.e. the curvature, and at the same time, each vortex/spin is attached by a flux of $U(1)$ gauge fields, i.e. the magnetic field. Notice that the former statement of flux attachment is equivalent to the usual particle-vortex duality (see Section 5), while the latter gives topological responses in the particle picture. To see it, we note that the Hall viscosity is equal to the spin density through $\eta_H = \frac{1}{2}s_0$ [62,75], where $s_0 \equiv \delta S/\delta \omega_0 = -CB$.[9] Therefore, the Wen-Zee term nontrivially relates the particle-vortex duality to the topological responses with the same coefficient $C$.

---

[9]Quickly, we have $\omega_0 \approx \frac{1}{2}\epsilon^{ab} e^i_a \partial_t e_{ib}$ where we used $e^0_\mu = \delta^0_\mu$ and $\Gamma^\sigma_{\rho 0} = 0$, so it gives a nonzero Hall viscosity $\eta_H = \frac{1}{2}\frac{\delta S}{\delta \partial_t e^a_i}\epsilon^{ab} e_{bi}$.

Now, let us look at (85). First, the shift of the dipole density is given by

$$\frac{\delta S_{CS}}{\delta A_0^a} = C\epsilon_{ab}\epsilon^{ij}R_{ij}^b(P). \tag{86}$$

The flux of the torsion field strength $\epsilon^{ij}R_{ij}^b(P)$ corresponds to the dislocation density, which also corresponds to the defects of translational symmetry [74]. In the meantime, the shift to the charge density is again given by

$$\frac{\delta S_{CS}}{\delta A_0} = -C\epsilon^{ij}\partial_i\omega_j. \tag{87}$$

These give rise to the fracton-elasticity duality proposed by [48]:[10] (86) gives rise to the dipole-dislocation duality, where the dipole gets attached by torsion flux, which corresponds to dislocation in elasticity theory; (87) gives rise to the charge-disclination duality, where the charge gets attached by curvature flux, which corresponds to disclination in elasticity theory. Moreover, the same coefficient $C$ in (86) and (87) implies that a dislocation is a bound state of two equal and opposite disclinations in accordance with [48].

Next, like the conventional Wen-Zee term, (85) also generates topological responses. We can define a spin current

$$\frac{\delta S_{CS}}{\delta\omega_\mu} = C\epsilon^{\mu\nu\rho}A_\nu^a e_\rho^a - C\epsilon^{\mu\nu\rho}\partial_\nu A_\rho, \tag{88}$$

and a stress-tensor

$$\frac{\delta S_{CS}}{\delta e_\mu^a} = -C\epsilon_{ab}\epsilon^{\mu\nu\rho}R_{\nu\rho}^b(D). \tag{89}$$

Working in flat spacetime afterward, we find a Hall viscosity

$$\eta_{\mathrm{H}} = \frac{1}{2}s_0 = \frac{1}{2}\frac{\delta S_{CS}}{\delta\omega_0} = \frac{C}{2}\left(\epsilon^{ij}A_i^a\delta_{aj} - \epsilon^{ij}\partial_i A_j\right), \tag{90}$$

a stress density

$$\rho_{\mathrm{stress},a} = \frac{\delta S_{CS}}{\delta e_0^a} = -C\epsilon_{ab}\epsilon^{ij}\partial_i A_j^b, \tag{91}$$

and a Hall elasticity

$$K_{\mathrm{H}} = \frac{1}{2}\frac{\delta S_{CS}}{\delta e_i^a}\epsilon^{ab}e_{bi} = -\frac{C}{2}\delta_{ci}\epsilon^{i\nu\rho}\partial_\nu A_\rho^c. \tag{92}$$

Both stress density and Hall elasticity are new topological responses absent in the conventional quantum Hall state, and they are present in the dipolar quantum Hall state due to the translational defects, i.e. dislocations. Defining an effective $U(1)$ magnetic field $B_{\mathrm{eff}} = \epsilon^{ij}\partial_i A_j - \epsilon^{ij}A_i^a\delta_{aj}$, a dipole magnetic field $B_{\mathrm{dip}}^b = \epsilon^{ij}\partial_i A_j^b$ and a dipole electric field $E_{\mathrm{dip}}^{ic} = \epsilon^{i\nu\rho}\partial_\nu A_\rho^c$, we can rewrite the Hall viscosity, stress density, and Hall elasticity as[11]

$$\eta_{\mathrm{H}} = -\frac{C}{2}B_{\mathrm{eff}}, \tag{93a}$$

$$\rho_{\mathrm{stress},a} = -C\epsilon_{ab}B_{\mathrm{dip}}^b, \tag{93b}$$

$$K_{\mathrm{H}} = -\frac{C}{2}E_{\mathrm{dip}}^{ic}\delta_{ic}. \tag{93c}$$

---

[10]We thank Leo Radzihovsky for discussions about it.

[11]Ideally, one wish to have the background fields being constant in order for the responses to be well-defined. This would require dipole and $U(1)$ gauge fields to be linear and quadratic in coordinates, respectively.

Hence, the effective $U(1)$ magnetic flux generates the Hall viscosity, the dipole magnetic flux generates stress density, and the trace of dipole electric field generates the Hall elasticity.[12] These relations go beyond the fracton-elasticity duality. In fact, this justifies calling (85) the dipolar Wen-Zee term since it nontrivially relates the fracton-elasticity duality to the topological responses with the same coefficient $C$.

One may wonder whether the dipolar Wen-Zee term (85) would lead to a boundary anomaly. In the case of a single $U(1)$ Wen-Zee term, the gauge transformation on a manifold with a boundary can be canceled by a local boundary counterterm [77], which implies that the $U(1)$ Wen-Zee term does not necessitate boundary anomaly. To see it, we introduce the embedding function $X^\mu(\sigma^A)$ where $\sigma^A$ are coordinates on $\partial M^3$. Then, we can define the projection $P[e^a] = e_A^a d\sigma^A$ by the pullback $e_A^a = \partial_A X^\mu e_\mu^a$. The local counterterm is given by the extrinsic one-form curvature $K$[13] at the boundary satisfying $P[\omega] + K = d\Phi$ where $\Phi$ is a boundary zero-form, such that $\int_{M^3} A \wedge d\omega + \int_{\partial M^3} A \wedge K$ is $U(1)$ gauge invariant. However, we find a similar calculation does not carry over to the action (85). Consider a dipole gauge transformation given by $\Lambda = \xi^a D_a$, then $\delta A^a = d\xi^a + \epsilon^{ac} \omega \xi^c$ and $\delta A = e^a \xi^a$, thus the action changes by

$$
\begin{aligned}
\delta S_{CS} &= \int_{M^3} \epsilon_{ab} d\xi^a \wedge de^b + \xi^a \omega \wedge de^b - \xi^a e^a \wedge d\omega - d\xi^a \wedge \omega \wedge e^a \\
&= \int_{M^3} d\left(\epsilon_{ab} \xi^a de^b - \xi^a \omega \wedge e^a\right) \\
&= \int_{\partial M^3} \epsilon_{ab} \xi^a \wedge P[R^b(P)].
\end{aligned}
\tag{94}
$$

Observing that $P[R^a(P)] = dP[e^a] + \epsilon_{ab} P[\omega] \wedge P[e^b] = dP[e^a] + \epsilon_{ab}(d\Phi - K) \wedge P[e^b]$, we realize that there exists *no* local counterterms that can cancel (94). Therefore, in a generic curved spacetime, the dipolar Wen-Zee action (85) contains a boundary anomaly. However, in certain special manifolds, (94) could be canceled by counterterms. One such example is when both the intrinsic and the extrinsic curvature vanish, i.e. $\omega = K = 0$, and, one can show that (94) can be canceled by the boundary term $\int_{\partial M^3} \epsilon_{ab} A^a \wedge P[e^b]$. As a remark, the boundary anomaly from (94) will lead to the violation of dipole conservation in the following way:

$$
\partial_A J_a^A = J^A e_{Aa} + \epsilon_{ab} \epsilon^{AB} R_{AB}^b(P).
\tag{95}
$$

Finally, let us come back to the 3D gravity interpretation of (85). It is suggested that such 3D gravity can be helpful for the study of 2D non-relativistic field theory [44] through holographic duality [78]. For our case, the dipole algebra (67) was recently realized as the infinite mass limit of the Galilean algebra [19]. Therefore, we hope that the 3D gravity derived in (85) would be useful in studying the so-called "flat band" models [79], which corresponds to infinite single-particle mass, through a field theory perspective.

# 7 Outlook

In this paper, we have constructed various dipolar Chern-Simons theories to describe topological responses for systems that conserve dipole symmetry. Our construction highlights a subtle issue of the problem: There is no need to impose a Lagrangian multiplier to reduce the dipole gauge theory to a higher-rank gauge theory. As a consequence, only the highest multipole symmetry can support a Chern-Simons term and its corresponding 't Hooft anomaly.

---

[12]It is interesting to see the relation with odd crystals [76].

[13]The precise definition can be found in [77].

An important lesson from this effective field theory construction is how to couple dipole symmetry to curved spacetime. Chronologically, it takes us three steps to understand this structure. First, we should think of the two indices of dipole gauge field $A_\mu^a$ as playing different roles under symmetries, in contrast to symmetric tensor gauge fields $A_{ij}$ of which the two indices are equivalent. In particular, the internal index $a$ indicates that the dipole gauge field transforms as a vector under rotational symmetry, and the spacetime index $\mu$ implies that it behaves as a 1-form under diffeomorphism. This structure significantly simplifies the construction of the dipolar Chern-Simons theory and allows us to see the effect of discrete rotational symmetry in unconventional bulk-edge correspondence. Second, our dipole gauge theory is in fact analogous to the vielbein $e_\mu^a$ that is used to gauge the spacetime symmetries [54]. Notice that on the one hand, the momentum is a time-reversal-odd vector charge, and the stress tensor is sourced by the vielbein; on the other hand, the dipole is a vector charge and its dipole current is sourced by the dipole gauge fields. This analogy carries over to the Chern-Simons theory. As we have shown, the torsional (vielbein) Chern-Simons theory can be similarly constructed using our approach. Moreover, this analogy promotes the third step of developing a consistent field theory on curved spacetime. Specifically, we should treat the dipole symmetry and the spacetime symmetry on an equal footing. This results in a non-abelian group symmetry. We emphasize that this non-abelian group symmetry is so far an unknown feature in higher-rank gauge theory, based on which people argued that dipole symmetry is inconsistent with a generic curved spacetime [34, 35]. As we have shown, dipole symmetry *can* survive on curved spacetime so long as there are dipole Goldstones or defects in geometry that could support it. Importantly, a careful analysis of coupling dipole symmetry to defects in geometry by a non-abelian Chern-Simons theory gives rise to a more comprehensive understanding of the fracton-elasticity duality [48].

Looking forward, we anticipate that our method can be generalized to subsystem symmetries. Recent work [56] studied the boundary anomaly with both continuous and discrete subsystem symmetries, but only adopt the perspective from boundary anomalous theory to bulk Chern-Simons theory. Understanding the rotational symmetry for those gauge fields will help to build the bulk theory directly and facilitate the analysis of bulk-edge correspondence. We expect our boundary-dependent bulk-edge correspondence would potentially extend the classification of SPT phases. In the meantime, it is possible to compare our theory to the one obtained by integrating out fermions in some microscopic dipole-symmetric fermionic models, possibly generalizing [80]. It is also interesting to quantize the theory in Section 6.2 following [43, 46], so that a topological quantum field theory or quantum gravity with dipole symmetry can be established.

## Acknowledgments

I acknowledge useful discussions with Leo Radzihovsky and Yizhi You. I especially thank Andrew Lucas for carefully reading the manuscript.

**Funding information** This work was supported by the Gordon and Betty Moore Foundation's EPiQS Initiative via Grants GBMF10279.

## A  Level quantization in the dipolar Chern-Simons theory

We derive the level quantization for the dipolar Chern-Simons theory. Similar derivation was done in [31].

For the purpose of this section, we take the dipole symmetry to be a compact $U(1)$ symmetry for the corresponding dipole moment living in the internal space. Imaging adiabatically moving a dipole moment pointing at $\hat{r}^a$ along a closed loop in the presence of the dipole gauge fields. Due to the single-particle dynamics governed by $\int \mathrm{d}t \; \partial_t x^i A_i^a \hat{r}^a$, the relative phase it picks up is

$$\alpha^a = \oint A_i^a \mathrm{d}x^i \, . \tag{A.1}$$

Now, threading a magnetic dipole flux through a sphere, and requiring consistency on the relative phase, the minimum such flux is given by

$$\int_{\mathbb{S}^2} B^a = 2\pi \hat{r}^a \, , \tag{A.2}$$

where $B^a = \epsilon^{ij} \partial_i A_j^a$, and $\hat{r}^a$ indicates the direction of the magnetic dipole moment.

Consider the thermal partition function $Z[A_\mu^a] = \exp(iS[A_\mu^a])$ of (2) with isotropic coupling $f_{ab} = \delta_{ab}$. Let the Euclidean time be periodic, $\tau \sim \tau + \beta$, and take the large dipole gauge transformation $\xi^a = -\frac{2\pi\tau}{\beta}\hat{r}^a$, the temporal dipole gauge field transforms as

$$A_0^a \to A_0^a + \frac{2\pi}{\beta}\hat{r}^a \, , \tag{A.3}$$

while $A_i^a$ remains invariant. Now, take $A_0^a$ to be constant, and thread the minimum flux to a sphere, (2) becomes

$$S = 4\pi C_2 \beta A_0^a \hat{r}^a \, . \tag{A.4}$$

Thus, under the large dipole gauge transformation, the action changes by

$$\delta S = 8\pi^2 C_2 \, . \tag{A.5}$$

In order for the partition function to remain the same, we must have

$$C_2 = \frac{k}{4\pi} \, , \quad k \in \mathbb{Z} \, . \tag{A.6}$$

This is the quantized level for dipolar Chern-Simons theory. To generalize to anisotropic cases, we should just require that the single-particle dynamics obey $\int \mathrm{d}t \; \partial_t x^i A_i^b \hat{r}^a f_{ab}$, so the relative phase it picks up becomes $\alpha_a = \oint f_{ab} A_i^b \mathrm{d}x^i$, which leads to flux $\int_{\mathbb{S}^2} B^b f_{ab} = 2\pi \hat{r}^a$. For the partition function to be invariant under the large dipole gauge transformation, we get the same condition as in (A.6).

To see the quantization of (18), one can imagine that the dipole moment is moving on a closed two-dimensional manifold. It will pick up a phase

$$\alpha_a = \oint A^b \wedge \delta^c f_{abc} \, , \tag{A.7}$$

where we used the differential forms $A^b = A_\mu^b \mathrm{d}x^\mu$ and $\delta^c = \delta_\mu^c \mathrm{d}x^\mu$ but assumed flat spacetime. This will lead to the minimum flux on a 3-sphere as

$$\int_{\mathbb{S}^3} \epsilon^{ijc} \partial_i A_j^b f_{abc} = 2\pi \hat{r}^a \, . \tag{A.8}$$

Similarly, in order for the partition function to be invariant under the large dipole gauge transformation, we must have

$$C_3 = \frac{k}{4\pi} \, , \quad k \in \mathbb{Z} \, . \tag{A.9}$$

Several remarks follow. First, one can further couple the theory to additional dynamical gauge fields to obtain fractional numbers in the coefficient mimicking the fractional quantum Hall effect. Second, the level quantization requries the symmetry to be compact. In Section 6.2, both the dipole symmetry and the translational symmetry are noncompact, so the level there is in general not quantized.

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
