# Peer review of "A Chern-Simons theory for dipole symmetry"

_SciPost Physics, doi:SciPost Phys. 15, 153 (2023)_

## Round 1 · Referee Report · Anonymous (Referee 1) · 2023-8-3

Strengths

1- Proposes a new field-theoretic avenue to study multipole symmetries and fracton-like physics;

Weaknesses

1- See requested changes

Report

The goal of the paper is to study fracton-like physics from the perspective of a field theory approach. It proposes an alternative path to construct field theories with exotic symmetries that are more systematic than the higher-rank field theories constructions. The main contribution of the work (in my perspective, please correct me if this is not the case) is the idea that the gauge fields associated with higher multipole symmetries must carry an additional internal index, that is associated with an internal spacetime. In this setting, the gauge fields are standard 1-form fields regarding the physical spacetime while being a rank N tensor regarding the internal spacetime.

Overall, I believe that the work is good and it is a nice addition to the literature and certainly deserves publication. But, first I would like that the author could address a few questions that I had while reading the paper and give some clarification in certain points.

Requested changes

1- around the equations (2.1a) and (2.1b), the author says that the fields $\phi$ and $\phi^x$ are compact and hence, have the appropriate U(1) transformations. The way I understand this is, if these fields are compact, then they must be functions on the circle $S^1$ (hence, the fields are maps from R to $S^1$). In (2.1b) this would require $\xi^x\in S^1$, but then this renders the term $x\,\xi^x$ in (2.1a) ill-defined, since $x\in R$ and $\xi^x \in S^1$. Am I missing something here?

2-the $C_{2n}$ constant first appears at eq.(1.2) but is not defined. It is worth defining it right below the equation (1.2). Also, regarding its quantization, how does my point in question (1) affect the level quantization?

3- Around (2.3), the author promotes the gauge functions $\alpha$ and $\xi^x$ to be local functions of the spacetime (gauging procedure), and then claims that "....to leading order ..." and writes down the equation (2.3). But, leading order in what? Is there some sort of perturbative argument playing a role here? Or are we simply throwing away pure gauge terms?

4-What are the counter-terms that lead to 2.5?

5-what kind of approximation is being made to obtain eq.(2.12)?

6- How does work dimensional analysis in this kind of theory? Does the internal spacetime contribute in any way? Or should we take the dimension of the fields regarding only the physical spacetime?

  • validity: high
  • significance: high
  • originality: high
  • clarity: good
  • formatting: good
  • grammar: excellent

Author:  Xiaoyang Huang  on 2023-08-15  [id 3905]

(in reply to Report 1 on 2023-08-03)

Request changes

1- What I really mean here is to consider a matter field $\Phi = e^{i \phi}$ that transforms under the dipole symmetry as $\Phi \to e^{i x \xi^x}\Phi$. So, in terms of the compact field $\phi$, the transformation is given by (2.1a). I include (mod $2\pi$) to emphasize the field $\phi$ is compact.

2- I add the definition of $C_{2n}$ below (1.2). The derivation of level quantization is based on the fact that the dipole symmetry is taken to be compact -- it is a U(1) symmetry for the corresponding dipole moment. The compact dipole symmetry can be constructed on a lattice as done in ref.[42]. On the other hand, if we take the dipole symmetry to be $\mathbb{R}$ for being a spacetime symmetry as discussed in Sec.6 (see also ref.[19]), then the quantization does not apply due to the non-compactness. However, whether or not the coefficient is quantized does not change our construction. In the meantime, the quantization is not relevant to the charge fields the referee mentioned in Q.1 because it is for Chern-Simons terms consisting of \emph{purely} dipole gauge fields. Indeed, at the end of Sec.2, we argued that Chern-Simons terms involving the U(1) charge gauge field are forbidden in dipole symmetric theories.

3-I modify the sentence to be ``to the leading order in $\alpha,\xi^x$''. This is the standard Noether current procedure, which is probed by infinitesimal transformations.

4-The second line of (2.5) is the counterterm. The first line, on the other hand, is the standard gauging procedure, i.e. the minimal coupling, where currents are coupled to the corresponding background gauge fields.

5-One way to see (2.13) is to note that if we shift $\phi^x\to \phi^{x\prime} = \phi^x + \partial_x\phi$, the $K_n$-term in the action generates a mass term for $\phi^{x\prime} $, so we can set $\phi^{x\prime} =0$, which gives (2.13). Equivalently, the EOM (2.12b) demands that in the long wavelength and late time limit, $J_x = 0$, which becomes the constraint (2.13). Certainly, (2.13) is valid to the leading order in derivatives. I expand the discussion around (2.13) to emphasize this point.

6-Indeed, the dimensional analysis should be taken with respect to the physical spacetime only. For example, we have $[A^a_\mu]=[A_\mu]=L^{-1}$ with $L$ the length. Notice that we also have $[\delta_\mu^a] = L^{-1}$ and $[x^a]=L^0$. The former is because in the flat spacetime, $e^a_\mu = \delta^a_\mu$ and $[e^a_\mu] = L^{-1}$; the latter is because $x^a$ lives in the internal spacetime. Then, we see both (1.1) and (2.1) have consistent dimensions, and, at the same time, the action (1.2) is dimensionless. I add a discussion below (2.9) along with a comparison to the higher-rank gauge theory approach.

---

## Round 1 · Referee Report · Anonymous (Referee 2) · 2023-8-6

Strengths

  1. The suggested formulation of Chern-Simons theory with both U(1) and dipole gauge fields is simple. The advantages of this approach vs. higher-rank gauge theory are clearly explained.

  2. The proposed formulation is well-suited for studying

  3. t'Hooft anomalies.
  4. boundary-bulk correspondence. In particular, it allows to describe boundaries in anisotropic systems naturally.
  5. particle-vortex duality for dipole symmetry as a straightforward generalization of the conventional one.

  6. Two ways of coupling the dipolar matter to gravity are proposed.

Weaknesses

  1. There are a few confusing points in the presentation, often due to grammar (I list some of them in the list of requested changes)

  2. The connection of the proposed dipole gauge theory to the higher-rank gauge theories used previously is not clearly explained. It would be useful for the reader to have the relationship between the two approaches. Are they equivalent or produce different results?

Report

I think that the manuscript meets the publication criteria for the SciPost Physics journal. The manuscript is well written, with the exception of minor points listed in the "requested changes" section. I propose publishing the manuscript in SciPost Physics after the minor revision.

Requested changes

  1. The second line of the introduction, "characterized by fracton," does not sound good. It would be better to replace it with something like "characterized by fractons - excitations with restricted mobility" or similar.

  2. Footnote 3 on page 3 is very confusing (grammar-wise). It is not clear what "it" means in "it sometimes...". Also, the reference [19], is it the reference where the "wrong conclusion" has been made or the one which explained the issue?

  3. The sentence "absorbing dipoles from the condensate" in the introduction section leads to more confusion than explaining anything.

  4. The last sentence of the introduction. "is" -> "are."

  5. On page 12, the first paragraph, "couple the dipolar Chern-Simons theory with matters" and "without coupling to matters," does not sound correct. Better to have "matters" -> "matter fields." In the first sentence, it would be nice to explain that "matter fields" mean dipole Goldstone field introduced below.

  6. The part of section 6 from the beginning to the beginning of subsection 6.1 repeats very closely (even verbatim) the Appendix B of the paper [19]. Moreover, some clarity has been lost in the attempt to shorten it compared to Appendix B of [19]. For example, the sentence "To make the vielbein matrix invertible, we define ..." does not make sense. Also, "we treat the vielbein and the spin connection as independent fields" is meant (see [19]) for variational purposes. I would suggest rewriting this part of section 6 for better clarity.

  7. In the beginning of 6.1 "The non-invariance... is structured" is confusing. What is "structured non-invariance"? Some explanation should be made.

  8. In 6.1, in "due to the kinetic constraint, it must be generated out of the condensate," it is not clear what "it" refers to.

  9. The title of Appendix A is very confusing. The Appendix is not about the "quantization of the dipolar Chern-Simons theory" but about the quantization of the level k of that theory. Similarly, "The Dirac quantization" usually refers to the quantization of theories with constraints. What the author means is "Dirac's argument for quantization of coupling constants." I suggest changing the title of Appendix A respectively and fixing the terminology to avoid such confusion.

  • validity: top
  • significance: high
  • originality: high
  • clarity: high
  • formatting: excellent
  • grammar: good

Author:  Xiaoyang Huang  on 2023-08-15  [id 3906]

(in reply to Report 2 on 2023-08-06)

Weakness

2-The higher-rank gauge theory is a result of a special gauge fixing of our dipole gauge field as already being discussed below (1.1). To make the comparison more concrete,
I add a discussion about the higher-rank gauge theory approach below (2.9)

Request changes

All the requests are fixed

---

## Round 1 · Referee Report · Anonymous (Referee 3) · 2023-8-11

Strengths

  1. It is overall well-written and easy to follow.
  2. It is also pedagogical for the readers, containing enough introductions and enough details on algebra.
  3. As the author correctly pointed out, writing down the Chern-Simons theory for the dipole symmetries, in particular using the rank-2 gauge fields, has been not entirely trivial. The author showed a clear recipe for writing it down, and also provided the generalizations to the higher dimensions.
  4. It also contains detailed studies on the interesting properties of the Chern-Simons theory and highlights well the difference from the original U(1) CS theory. A new type of vortex-particle duality was also interesting and probably attracts some more future studies.
  5. Gravitational responses (including the coupling to the background geometry) are also carefully handled.

Weaknesses

  1. Based on the field theory, the contents are likely not entirely transparent to most condensed matter theorists, who are not familiar with these types of field theories.
  2. Not much of a physical picture or intuitions has been provided. E.g. some central physics of quantum Hall states can be easily captured from the Laughlin's argument or pumping. Such a picture (or trial of explaining in such a fashion) will greatly help to attract a broader audience.

Report

The manuscript by Huang discusses the Chern-Simons theory for multipoles, which advances the field in several aspects (when compared to the previous works). Unlike the previous works which relies on the higher-rank gauge fields, this paper utilizes the two U(1) gauge fields, A and A^a, a=x,y,z, which help in writing down the Chern-Simons theory in a quite clear manner. In particular, its anomaly is nicely matched by lower-dimensional chiral boson models. Furthermore, this approach allows to couple the theory to the background geometry. Overall, I strongly recommend accepting the manuscript to the journal.

Nonetheless, I have a few questions, which I put in the "requested changes" below, which I hope the author to reflect in the manuscript.

Requested changes

  1. Is the dipolar Chern-Simons theory really the theory of a SPT? I can see what the author tries to say in the part where the author connects the physics to the SPT (section 3.3), however, when people mention SPT, it means that the boundary is not gappable when proper symmetries are imposed. In this part, can the author show (1) that the boundary is gappable (for the boundary without the symmetries) and (2) that the boundary mass terms are prohibited for the symmetric boundaries?

  2. Can the author discusses the breaking of gauge fields to the discrete subgroups, such as Z2?

  3. In the appendix, the author showed the quantization of the dipolar Chern-Simons theory. Does it depend on the statistics of the dipole, perhaps if it's fermionic or bosonic? Or it doesn't matter?

  • validity: high
  • significance: high
  • originality: high
  • clarity: high
  • formatting: good
  • grammar: excellent

Author:  Xiaoyang Huang  on 2023-08-15  [id 3907]

(in reply to Report 3 on 2023-08-11)

Weakness

2-I think, to some extent, the Laughlin's argument shares the same physics as doing large gauge transformation in Chern-Simons theory as done in appendix.A. I guess the physical intuition that I am trying to provide is to take a direct analogy between the dipole CS theory and the usual charge CS theory.

Requested changes

1-I agree to the referee that the ``SPT'' usually refers to a theory where the boundary is gappable in the absence of the symmetry. Here, however, the boundary is \emph{always} gapless even if the symmetry we care about is broken. Specifically, the symmetry that corresponds to the SPT phase is the parity normal to the boundary, and within the SPT phase, the boundary mode is quadratic. Further, I explicitly constructed cases where this parity symmetry is broken, and I found that the boundary modes become one linear and one cubic dispersing. Now, similar to the conventional gapped boundary mode, the symmetry broken phase cannot be adiabatically connected to the SPT phase because as we see in (3.10) and (3.25), the $\theta\to 0$ limit involves a singularity. I am not aware of a similar case in the literature (probably due to my ignorance), so I think SPT is a good jargon for it.

2- I agree in general it is possible to generalize our construction to discrete gauge field and to subsystem symmetry following ref.[56], however, I think that requires extensive works and is better to be a separate paper.

3-The (dipolar) Chern-Simons theory is a theory for gauge fields and does not couple to matter fields. The level quantization is purely a fact that a topologically non-trivial gauge transformation is allowed. The appendix.A follows closely to this argument so the particle statistics are not relevant.

---

## Editorial Decision

published